# MACC1-Induced Collective Migration Is Promoted by Proliferation Rather Than Single Cell Biomechanics

**DOI:** 10.3390/cancers14122857

**Published:** 2022-06-09

**Authors:** Tim Hohmann, Urszula Hohmann, Mathias Dahlmann, Dennis Kobelt, Ulrike Stein, Faramarz Dehghani

**Affiliations:** 1Department of Anatomy and Cell Biology, Martin Luther University Halle-Wittenberg, Grosse Steinstrasse 52, D-06108 Halle (Saale), Germany; tim.hohmann@medizin.uni-halle.de (T.H.); urszula.hohmann@medizin.uni-halle.de (U.H.); 2Experimental and Clinical Research Center, Max-Delbrück-Center for Molecular Medicine in the Helmholtz Association, Charité—Universitätsmedizin Berlin, Robert-Rössle-Straße 10, D-13125 Berlin, Germany; mathias@kaddea.de (M.D.); dennis.kobelt@epo-berlin.com (D.K.); 3German Cancer Consortium (DKTK), Im Neuenheimer Feld 280, D-69120 Heidelberg, Germany

**Keywords:** MACC1, migration, biomechanics, proliferation, colorectal cancer

## Abstract

**Simple Summary:**

High metastasis-associated in colon cancer 1 (MACC1) expression is associated with metastasis, tumor cell migration, and increased proliferation in colorectal cancer. Tumors with high MACC1 expression show a worse prognosis and higher invasion into neighboring structures. However, the mediation of the pro-migratory effects is still a matter of investigation. The aim of this study was to elucidate the impact of single cell biomechanics and proliferation on MACC1-dependent migration. We found that MACC1 expression associated with increased collective migration, caused by increased proliferation, and no changes in single cell biomechanics. Thus, targeting proliferation in high-MACC1-expressing tumors may offer additional effects on cell migration.

**Abstract:**

Metastasis-associated in colon cancer 1 (MACC1) is a marker for metastasis, tumor cell migration, and increased proliferation in colorectal cancer (CRC). Tumors with high MACC1 expression show a worse prognosis and higher invasion into neighboring structures. Yet, many facets of the pro-migratory effects are not fully understood. Atomic force microscopy and single cell live imaging were used to quantify biomechanical and migratory properties in low- and high-MACC1-expressing CRC cells. Furthermore, collective migration and expansion of small, cohesive cell colonies were analyzed using live cell imaging and particle image velocimetry. Lastly, the impact of proliferation on collective migration was determined by inhibition of proliferation using mitomycin. MACC1 did not affect elasticity, cortex tension, and single cell migration of CRC cells but promoted collective migration and colony expansion in vitro. Measurements of the local velocities in the dense cell layers revealed proliferation events as regions of high local speeds. Inhibition of proliferation via mitomycin abrogated the MACC1-associated effects on the collective migration speeds. A simple simulation revealed that the expansion of cell clusters without proliferation appeared to be determined mostly by single cell properties. MACC1 overexpression does not influence single cell biomechanics and migration but only collective migration in a proliferation-dependent manner. Thus, targeting proliferation in high-MACC1-expressing tumors may offer additional effects on cell migration.

## 1. Introduction

Colorectal cancer (CRC) is the third most frequent cancer type and a major cause of cancer-associated death [1]. The survival of patients is directly linked to the formation of metastasis as the most frequent type of therapy failure [2,3]. For the formation of metastasis, tumor cells need to detach from the primary tumor and migrate either individually or collectively through surrounding tissues and enter vessels.

In recent years, metastasis-associated colon cancer 1 (MACC1) has been shown to promote migration and proliferation in CRC and be a predictor of clinical outcome [4,5]. High *MACC1* expression levels, whether determined in the primary tumor or in the cancer patient’s blood, predict a reduced survival time, caused by a promotion of tumor aggressiveness and metastasis formation, even when detected in early, not yet metastasized tumor stages [6]. MACC1 has been established by us and many other groups as crucial for tumor development and metastasis. Thus, it has been found to be a prognostic and predictive biomarker for tumor progression and metastasis formation in a multitude of solid cancer types, including supportive meta-analyses for solid cancers; hepatocellular and gastrointestinal tract cancers, such as CRC and gastric cancer; and gynecological and breast cancer [7,8,9,10,11,12].

MACC1 regulates key processes during tumor progression such as proliferation, migration, invasiveness, and metastasis formation in xenografted and transgenic mice [6,13]. Cell migration represents a very crucial phenomenon and reflects, in addition to cell proliferation, a most decisive feature. However, how MACC1 is mechanistically increasing and thereby decisively activating cell migration is not well understood.

Thus, the current study investigated MACC1-induced elevated cancer cell motility, focusing on the biomechanics and migration of CRC cells on the single and multicellular level. Furthermore, the effect of proliferation on collective cell migration was assessed. We thereby found that MACC1 expression does not affect the migration and biomechanics of single colorectal cancer cells but does affect collective migration dynamics in a proliferation-dependent manner.

## 2. Materials and Methods

### 2.1. Cell Culture

SW480 and SW620 cells were purchased from the American Type Culture Collection (Manassas, VA, USA). The generation of the MACC1-overexpressing cell line SW480/MACC1 and MACC1-silenced SW620/shMACC1 and their respective controls (SW480/EV and SW620/EV) was previously described [4].

SW620 cells were cultured in 89 % (*v*/*v*) DMEM (Invitrogen, Carlsbad, CA, USA 41965-062) and SW480 in 89% (*v*/*v*) RPMI (Lonza, Basel, Switzerland, BE12-115F) and both were supplemented with 10 % (*v*/*v*) FBS (Gibco, Life Technologies, Carlsbad, CA, USA, 10500-064) and 1% (*v*/*v*) penicillin/streptomycin (Gibco, 15140-122).

To inhibit cell proliferation, 0.1 µg/mL mitomycin was applied 1 h before the start of the respective experiment.

### 2.2. Quantitative Real-Time Reverse Transcription PCR (qRT-PCR)

Total RNA was isolated using a GeneMATRIX Universal RNA Purification Kit (Roboklon, Berlin, Germany) according to the manufacturer’s instructions. In total, 50 ng RNA was taken from each sample for reverse transcription (RT) with random hexamers in a reaction mix (25 μM hexamer primer, 200 U/μL reverse transcriptase, 40 U/μL RNase inhibitor, 5× synthesis buffer, dNTP mix, PCR grade water; all from Biozym, Hessisch Oldendorf, Germany). RT was carried out at 30 °C for 10 min, 50 °C for 40 min, and 99 °C for 5 min with cooling at 4 °C for 5 min and samples were diluted 1:1 with PCR grade water. qRT-PCR was performed in duplicate with the Blue SYBR Green qPCR Mix (Biozym) in a final volume of 10 μL in a LightCycler 480 (Roche, Basel, Switzerland). The PCR protocol included a pre-incubation step at 95 °C for 2 min, followed by 40 cycles of incubation at 95 °C for 5 s, and annealing at 60 °C for 20 s. In the melting curve, the temperature was increased from 65 to 95 °C (0.1 °C/s). RNA polymerase II (RP II) was used as the housekeeping gene. The primers for MACC1 (fow 5′-ttcttttgattcctccggtga-3′; rev 5′-actctgatgggcatgtgctg-3′) and RP II (fow 5′-gaagatggtgatgggatttc-3′; rev 5′-gaaggtgaaggtcggagt-3′) were used for qPCR amplification of cDNA by a LightCycler 480 II (Roche Diagnostics, Basel, Switzerland). Data were analyzed with the LightCycler 480 Software release 1.5.0 SP3 (Roche Diagnostics). Average values of repeated samples were taken, and each mean value of the expressed genes was normalized according to the results of the housekeeping gene. All expression analyses were performed three times independently.

### 2.3. Western Blot Analysis

RIPA buffer (50 mM Tris-HCl at pH 7.5, 150 mM NaCl, 1% NP-40; supplemented with protein and phosphate inhibitor cocktail tablets, Roche Diagnostics) was used for cell lysis on ice for 30 min. Protein concentration was quantified with the Bicinchoninic Acid (BCA) Protein Assay (Thermo Scientific, Waltham, MA, USA).

Sodium dodecyl sulfate-polyacrylamide gel electrophoresis (SDS-PAGE) was used to analyze the protein expression levels. After electrophoresis, transfer to the polyvinylidene difluoride (PVDF) membrane was performed. Membranes were blocked with 5% milk in TBS-T (50 mM Tris-HCl, 150 mM NaCl, 0.05% Tween 20, pH 7.5) for 1 h at room temperature. Membranes were washed with TBS-T and treated with MACC1 or β-actin (as loading control) primary antibody (MACC1 antibody, Sigma, diluted 1:3000; β-actin antibody, Sigma, dilute 1:20,000; all prepared with albumin bovine fraction V; Serva, Heidelberg, Germany) at 4 °C room temperature overnight. The membranes were incubated with horseradish peroxidase (HRP)-conjugated secondary antibodies (anti-rabbit IgG, Promega, dilute 1:10,000; anti-mouse IgG, Thermo Fisher, dilute 1:40,000; prepared with TBS-T) at room temperature for 1 h, and washed. Detection was performed with WesternBright (Advansta, Menlo Park, CA, USA) and subsequent exposure to Fuji medical X-ray film SuperRX (Fujifilm, Tokyo, Japan). Quantification was performed using Image J 1.53a (National Institute of Health, Bethesda, MD, USA). 

### 2.4. Atomic Force Microscopy

The mechanical properties of single cells were assessed in the form of the Young’s modulus and cortex tension; both were measured using an atomic force microscope (AFM; Bruker, Billerica, MA, USA, Bioscope Catalyst). The measurement procedure for obtaining the Young’s modulus is described elsewhere [14,15]. Briefly, cells were seeded on a petri dish and measured 15 min after seeding to avoid slippage of individual cells. Measurements were conducted using a tip-less cantilever (Arrow-TL2, Nanoworld, Neuchatel, Switzerland) to apply a force of 1 nN that led to deformations of 1–2 µm. The Young’s modulus was calculated using the Hertz model:$$F=\frac{4}{3}\;\frac{E}{1-\nu^{2}}\;\sqrt{R\delta_{0}³}$$
where *F* denotes the applied force, *E* is the Young’s modulus, *R* is the cells radius, *ν* is the Poisson ratio (set to 0.5), and *δ*_0_ is the central indentation.

From the same measurement curves, the actin cortex tension was extracted. For this purpose, a model introduced by Cartagena-Rivera et al. [16,17] was used:T=kπ1Zd−1
where *T* is the cortex tension, *k* is the elastic constant of the cantilever, *Z* is the piezo extension, and *d* is the deflection of the cantilever. To calculate the surface tension, the first 200 nm of the force–distance curve after contact between the cantilever and cell were fitted. Please note that the difference in indentations (factor 5–10) allows a disentanglement between the cortical Young’s modulus and cortical tension. When fitting the force–distance curves for extraction of either the Young’s modulus or the cortex tension, values were discarded if R^2^ < 0.8.

### 2.5. Single Cell Motility and Doubling Time

For time lapse microscopy, 1000 cells were seeded in a 6-well plate 24 h before the start of the experiments. Cells were imaged with a microscope (Leica DMi8, Leica, Wetzlar, Germany) equipped with temperature (37 °C) and CO_2_ regulation (5% (*v*/*v*)). The experiments were conducted as described before [14,18]. From the obtained images, the contact area of the cells with the substrate, the mean speed, and the directionality were determined. Directionality was defined as the distance between the start and endpoint of a cell divided by the sum of incremental movements, and ranges between 0 and 1 for random or straight movements, respectively. To characterize the type of motion further, the Fürth formula was used to extract the persistence of motion and the diffusion coefficient from the mean squared displacement (MSD). The Fürth formula is defined as [19]:$$MSD\left(t\right)=4\times D\times t_{0}\;\left(e^{-t/t_{0}}+\frac{t}{t_{0}}-1\right)$$
where *t* is the time, *t*_0_ is the time of persistent movement, and *D* is the diffusion coefficient. As the images displayed cell divisions, the doubling time for each cell line was determined assuming exponential growth.

### 2.6. Measurement of the Properties of Collective Cell Migration

First, 24 h after seeding either 400,000 SW480 or 600,000 SW620 cells into 12 wells, they were transferred to an inverted microscope (DMi 8, Leica, Wetzlar, Germany) with temperature (37 °C) and CO_2_ (5% (*v*/*v*)) control. Images were taken every 5 min for 20 h and filtered using block matching 3D transform [20]. To analyze the local velocity, particle image velocimetry (PIV) [21,22,23] was used with a cross-correlation window size of 32 × 32 pixels (pixel size: 0.48 µm).

To quantify the cellular movement, the self-overlap function *Q*(Δ*t*) [24] was calculated:$$Q\left(∆t\right)=\frac{1}{N}\overset{N}{\underset{i=1}{\sum }}w_{i}\;\;\;with\;w=\left\{\begin{array}{c} 1;if\;\Delta r>0.1d\\ 0;else\;\;\;\;\;\;\;\;\;\;\;\;\;\;\;\;\;\;\; \end{array}\right.$$
where *N* is the cell number, *Δr* is the distance to the initial position of each cell, and *d* is the cell diameter. Here, *d* = 80 px (≈38.4 µm) was chosen for all cell types, corresponding approximately to the cell diameter of SW480 cells. *Q* measures the proportion of cells that moved away more than 10% of its cell size from their initial position. For quantification of the cooperativity, the 4-point susceptibility *χ* was calculated:$$\chi =N\left[\langle Q{\left(\Delta t\right)}^{2}\rangle -\langle Q{\left(\Delta t\right)\rangle }^{2}\right]$$

The peak height of *χ* is proportional to the number of cells moving collectively and the peak position corresponds to the average lifetime of collectively moving cell packs [24,25].

Furthermore, to assess caging for each non-boundary cell, how many of its eight nearest neighbors at the beginning and end of each measurement were identical was checked using an Euclidian distance metric.

### 2.7. Measuring Colony Expansion

As collective migration and expansion depends on cell density, the migration of small, growing cell clusters with a defined cell number was analyzed. Therefore, cells were seeded sparsely and allowed to form small clusters (4–12 cells) before imaging every 5 min for 40 h. From the measurements, the cellular speed inside the clusters, the cluster size in absolute terms and normalized to the cell number, the amount of cellular reorganization, and measures for directionality of the cellular movement (auto-correlation length, changes of neighborhood, angular variance of the velocity field) were determined. To characterize the growth of colonies further, the expansion speed $v\left(t\right)=dr\left(t\right)/dt$ of the cluster was calculated, taking advantage of the circular cluster geometry. Thus, it holds for the time differential of the cluster area *A*:$$\frac{dA\left(t\right)}{dt}=2\times \pi \times r\left(t\right)\times \frac{dr\left(t\right)}{d\left(t\right)}\;\;\to \;\frac{dr\left(t\right)}{d\left(t\right)}=\frac{dA\left(r\right)}{dt}/P\left(t\right)\;$$
with perimeter *P*. Please denote that the expansion speed *v* decreases with an increasing cluster size, given the constant area change. Notably, if *A* increases linearly, $dA\left(t\right)/dt=const.,$ it is easily possible to extrapolate the expansion rate of the cluster to a size corresponding to one cell size for comparison to the speed of single cells.

To determine the cell number in clusters of SW480 cells, the cell number in each cluster was counted at 0, 10, 20, 30, and 40 h and interpolated using an exponential fit. SW620 cells formed 3D clusters with no visible single cells. Therefore, the doubling time determined from the single cell measurements was used to estimate the single cell volume for each cluster, relative to the initial cell volume. It was calculated as the product of the projected area of the cluster and cluster intensity normalized to the background intensity, normalized to the value at the start of the measurement.

### 2.8. Simulation of a Persistent Random Walker with Directional Constraints

To model the expansion of small cell clusters, without proliferation as a constraint, the persistent random walk model was used. The movement of cells was modeled as: $∆\vec{x\left(t\right)}=v\times \left\{\begin{array}{c} \cos\left(\alpha \left(t\right)\right)\\ sin\left(\alpha \left(t\right)\right) \end{array}\right.$, with time *t*, spatial coordinates $\vec{x}$, speed *v*, and direction of movement *α*. To induce persistence, the following approach was used: $\alpha \left(t_{a}\right)=\left\{\begin{array}{c} \alpha \left(t_{a-1}\right)\;if\;P\geq \;R_{1}\;\\ 2\times \pi \times R_{2}\;if\;P<R_{1} \end{array}\right.$.

*P* is the persistence of movement, varying between 0 and 1, with 1 corresponding to linear movement and 0 to random movement. *R_1_* and *R_2_* are random numbers between 0 and 1. To obtain the persistence of movement, the results of the single cell motility analysis were used. To model boarder cells in small cell clusters, the same approach, together with the previously obtained persistence, was used, but the movement was set to zero if the modeled cell moved in the negative *x* direction: $∆\vec{x\left(t\right)}=0\;if\;\cos\left(\alpha \left(t\right)\right)<0$, as illustrated in Figure A1. This assumption was made to reflect the observation that cells did not migrate back into the cell cluster. As readout, the cellular displacement from its initial position per hour was determined, corresponding to the expansion speed of the cluster $dr\left(t\right)/dt$. For more details, see the Appendix B.

### 2.9. Statistics

Statistics was performed using the two-sided Mann–Whitney–Wilcoxon test or Kruskal–Wallis test with the Tukey post-hoc test. Significance was defined for *p* < 0.05. All error bars depict the standard error of the mean. Experiments were repeated at least three independent times. All sample sizes are summarized in Table A1.

## 3. Results

### 3.1. Effect of MACC1 Expression on Single Cell Properties

To assess the biomechanical properties of the two CRC cell lines SW480 and SW620 under conditions of low and high MACC1 expression, the Young’s modulus and cortical tension were measured. For SW480/EV (low endogenous MACC1 expression, Figure A2) and SW480/MACC1 (high endogenous MACC1 expression), the median modulus was 570 and 478 Pa, respectively, while it was significantly higher for the SW620/shMACC1 (low endogenous MACC1 expression, *p* < 0.0001) and SW620/shCTL (high endogenous MACC1 expression, *p* < 0.0001) cells (2044 and 2017 Pa, respectively; Figure 1A). No MACC1 dependence on the cortical tension was found, but SW480 cells (125 or 175 pn/µm) had significantly lower values than SW620 (356 or 272 pn/µm, *p* = 0.012 and *p* = 0.048, Figure 1B) cells. Next, it was evaluated whether single cell migration was affected by MACC1 expression. Thereby, we found that all cell types moved comparably fast, with 10–11 µm/h, independent of MACC1 expression (Figure 1C). Notably, the movement was highly undirected, with directionalities between 0.03 and 0.06 (Figure A3). Combining the average speed and directionality, it led to an average displacement of ≈7 µm/day for SW620 and ≈14 µm/day for SW480 cells. The average displacement for both cell types was less than one cell size, implying that single cells are rather stationary. These observations are in line with the fit results for the Fürth formula to characterize diffusion and persistence of movement. While the mean squared displacement was well approximated by the Fürth equation (R^2^ > 0.85), values for all cell lines were very similar, with diffusion coefficients in the order of 0.2 µm^2^/min and a persistence time of 5 min, which was the time difference between successive images. Furthermore, the contact area of SW480 cells with the substrate dropped from 2435 to 1530 µm^2^ upon MACC1 overexpression (*p* < 0.0001). Silencing MACC1 in SW620 increased the contact area from 817 to 949 µm^2^ (*p* = 0.025). From the live cell images, the doubling times were calculated, confirming the proliferative effect of MACC1 on these CRC cell lines. For SW480 cells, the doubling time decreased from 29 to 22 h upon MACC1 overexpression while silencing of MACC1 in SW620 cells increased the doubling time from 14 to 18 h.

### 3.2. MACC1 Promotes Collective Migration in a Cell Line-Dependent Manner

Next, it was analyzed whether MACC1-mediated pro-migratory effects necessitate cell–cell interactions. Therefore, collective migration of a cell layer was analyzed using particle image velocimetry. It was found that SW480/MACC1-expressing cells moved significantly faster (13–15 µm/h) than the cells with low MACC1 expression (8–9 µm/h; *p* < 0.0001, Figure 2A,B,D), but the same dependency for SW620 cells was not found upon MACC1 silencing (both: 4–6 µm/h; Figure 2C,E).

For further analysis of collective migration, it was measured how long cells needed to move from their initial location at least 0.1 cell diameters away. For SW480 cells, MACC1 overexpression resulted in cells moving away faster from their initial position (70 min vs. 180 min for 50% of the cells to move significantly, *p* < 0.0001, Figure 3A). This effect was not observed for SW620 cells upon MACC1 silencing (395 min vs. 430 min, Figure 3 B). Using the order parameter, the four-point susceptibility was calculated. The peak time of this quantity corresponds to the time a fast-moving pack of cells moved together. After several pack life times, the cell layer underwent significant reorganization. The reorganization times in SW480/MACC1 cells compared to SW480/EV were lower (65 min vs. 220 min; *p* < 0.0001, Figure 3C), but no effect was observed in SW620 cells (275 min for both; Figure 3D). As an additional independent metric for cell layer reorganization, we analyzed how many cells did not make contact with new neighbors during the measurement time of 20 h. It was found that 22% of SW480/EV cells but only 7% of SW480/MACC1 (*p* < 0.0001) cells did not make any new cell–cell contacts (Figure 3E) while both SW620 cell populations behaved similarly (71% and 70%, respectively, Figure 3F). Taken together, MACC1 overexpression induced a faster moving phenotype associated with faster reorganization of the monolayer of SW480 cells but not in SW620 cells. Notably, SW620 cells did not form a monolayer but rather a multilayered structure, where no individual cells could be distinguished. Thus, particle image velocimetry (PIV) detected the movement of the top cell layer interacting only with cells below and not with the substrate as is the case for SW480 cells.

The analysis of the velocity maps showed a strong association between proliferation events in both populations of SW480 cells and peaks in the velocity map (Figure 4). During mitotic rounding, the cell itself contracted and thus moved quickly and surrounding cells filled the free space, creating regions of high velocity. After cell division, daughter cells expanded, creating another region of high velocity, albeit with lower peak velocities. Taken together, with the lower doubling times of SW480/MACC1, we reasoned that cell division might be an important factor in MACC1-induced migratory effects.

### 3.3. MACC1 Promotes Colony Expansion and Migration Dependent on Proliferation

To analyze the effects of the cell division rate on the movement of small cell groups, the expansion and growth of small colonies of SW480 and SW620 cells (4–12 cells) were analyzed over the time course of 40 h (Figure 5A, Appendix A). This approach allowed the precise determination of cell numbers in each colony for SW480 cells at any time and thus helped to decouple cell division from migration. SW620 cells immediately formed dense 3D clusters that did not allow the discrimination of individual cells. Of note, in all analyzed clusters, no cell left the cluster. Analysis of the cellular movement speeds recaptured values obtained in the dense, large monolayer (Figure 5B–E) and MACC1-associated effects on speed (*p* = 0.011) were conserved. As clusters were comparably small and cells synchronized, cell divisions often occurred in quick succession, demonstrating that cell divisions can induce high velocities in single clusters (Figure A4).

Next, the cluster size was analyzed, depicting larger clusters for SW480/MACC1 cells, compared to SW480/EV cells (Figure 6A, *p* = 0.037), while SW620/shMACC1 and SW620/shCTL clusters were of a very similar size (Figure 6B). Cluster size increased exponentially for all cell types (R^2^ > 0.99) and the difference in the cluster size between SW480/EV and SW480/MACC1 increased over time. When normalized to the number of cells (SW480 cells) or relative cell volume (SW620 cells), the curve reflected the average cell size or volume and all previously observed effects diminished and the curves of the high- and low-MACC1-expressing cells run parallel. The average cell size decreased for all populations and thus the rate of outward migration did not keep up with the rate of proliferation (Figure 6C,D).

Furthermore, when analyzing the movement characteristics in terms of the changes in the cellular neighborhood, the autocorrelation of the velocity field inside the cell clusters, and the angular variance of the velocity fields, high- and low-MACC1-expressing cells showed only very little cellular reorganization and were otherwise indistinguishable (Figure A5).

To verify whether the MACC1-induced increase in cellular velocities was caused by proliferation, 0.1 µg/mL mitomycin was applied to the SW480 clusters immediately before the imaging to inhibit proliferation. As mitomycin effects are time delayed and it additionally causes apoptosis, the analysis window was restricted to the time of 10–26 h after mitomycin treatment, as no significant cell death or proliferation was observed in that period (Figure 7A). During that time window, SW480 cells in clusters of both populations had virtually the same speed of ≈3 µm/h over the whole time frame (Figure 7B,C), being significantly slower than in the untreated cell clusters (*p* < 0.0001). Furthermore, cluster expansion was now linear and—due to the constant number of cells in each cluster—only determined by the formation of protrusions of cells. SW480/EV (slope: 537 ± 2 µm^2^/h) cell clusters expanded faster than SW480/MACC1 (slope: 326 ± 1 µm^2^/h) clusters, independent of the cell number (Figure 7D,E). Extrapolated to the size of a single cell, cluster expansion rates were calculated for SW480/EV (*v* = 3.09 ± 0.05 µm/h) and SW480MACC1 (*v* = 2.36 ± 0.01 µm/h). Notably, these values were approximately four to five times higher than the values obtained for the displacement speed of single SW480 cells (14 µm/24 h ≈ 0.58 µm/h). At the border of cell clusters, the movement of SW480 cells was bound to outward movements because on the side of the cluster, other confined cells inhibited motion into the cluster. Consequently, the degrees of freedom for directional choices of migration were approximately halved. This assumption was in line with our observations showing that no boarder cell moved into the cluster during the measurements.

Assuming a persistent random walk for single cells, one can extract the persistence of the movement of SW480 cells using the average speed and displacement measured for single cells (see the Appendix A). The persistence parameter, bound to the values 0 to 1, was found to be *p* = 0.05, which is very low and in agreement with the low values of the persistence time reported. Using this persistence and performing the same analysis for cells at the boarder of the cluster, restricting movement in one direction, we obtained cluster displacement speeds of 3.18 ± 0.27 µm/h, which are close to the ones obtained from the experiment.

## 4. Discussion

In this study, we report MACC1-induced effects on the collective migration of CRC cells. It was demonstrated that these effects are not mediated via changes in single cell motility, elasticity, or cortex tension. Interestingly, MACC1-induced proliferation appeared to be an important inducer of the increased MACC1-dependent collective migration.

MACC1 has proven to be a prognostic marker that is predictive of therapy response and targetable by various drugs [4,5,26,27]. Two important effects associated with increased MACC1 expression are increased cell migration and proliferation [4,5], both being hallmarks of cancer. From a mechanistic point of view, MACC1 induces the activation of the HGF/c-Met axis in multiple tumor entities [4,5,28,29,30]. Another main component of MACC1 signaling is via PI3K/Akt [5,31,32]. Both the HGF/c-Met axis and PI3K are strongly involved in cytoskeletal reorganization and regulation of cell migration [33,34,35]. Consequently, MACC1 was demonstrated to induce increased cell migration [4,26,29,36,37,38,39,40]. Based on these studies, we expected an MACC1-induced increase in single cell and/or collective migration, yet we could verify this only for collective migration. Notably, in a previous study of ours, increased single cell velocities of glioblastoma cells were found upon increased MACC1 expression, which were associated with differential biomechanical properties and cytoskeletal organization [36]. As no changes in the biomechanics of single cells were observed here, it is plausible that single cell motility was unchanged. A further likely difference between the previous and the current study is the differential coupling of signaling cascades in both tumor entities, which may cause different downstream effects. On the other hand, SW480 and SW620 cells were stationary as isolated cells, potentially due to different cell surface friction compared to the glioblastoma cells [23,41], thus behaving highly differently from the beginning. Still, compared with the other studies, both mechanistic and functional, conducted in CRC, changes in migration are to be expected. Yet, these studies were mostly performed using Boyden chambers or scratch assays, measuring either dominant chemotaxis or a combination of proliferation and (collective) migration. To assess if MACC1-dependent effects necessitate cell–cell interactions, collective migration was analyzed. Thereby, in SW480 cells, MACC1 expression increased migration and promoted layer reorganization. In other studies, such changes were associated with changes in cell–surface interactions and reductions in cell–cell adhesion or their ratio [23,42], which is in line with the MACC1-dependent increase in fibronectin and decrease in E-cadherin expression [43,44]. Nevertheless, the experiments performed here imply either the necessity of cell–cell interactions or confinement for MACC1-induced migration. Notably, cell division events occurred in parallel with local peaks in the velocity field. In preparation for division, a cell contracts and thus locally reduces pushing forces on neighboring cells while simultaneously freeing up space. Thus, neighbors quickly expand, occupy the free space, and are displaced again when the two daughter cells start to expand [45]. Consequently, proliferation may induce tissue fluidization and thus migration [46,47] via the induction of active stress fluctuations [48,49,50,51].

To analyze the effect of proliferation further, the system was downscaled to small cell clusters, with a defined number of cells. Downscaling reproduced the key properties of the dense layer measurement: the mean speed inside the cell clusters and high velocities associated with proliferative events. In the presence of proliferation, the expansion of cell clusters was dominated by proliferation during the whole measurement time, as seen by the exponential growth of colonies and the decrease in the average cell size, as reported before [52,53]. Notably, the used cell types migrate too slowly to match proliferation in any state of cluster expansion. In contrast, in the absence of proliferation, the velocity inside of the clusters was smaller [54] and the cluster size increased linearly, solely dominated by cell migration [52]. This observation also agrees with a recent study showing that inhibition of proliferation is highly effective in arresting motility [47]. Devany et al. argued that the shape, force, and motility fluctuations may largely be caused by proliferation rather than other cellular processes classically associated with cell motility, such as cytoskeletal remodeling [47]. Given the low single cell motility of the cell types examined here, this argument appears plausible. While the model used to simulate the movement of SW480 cells in the absence of proliferation is over-simplified because it does not take into account any kind of cell–surface or cell–cell interactions, it gave a reasonable prediction of the outward movement of cells. Thus, cluster expansion in the absence of proliferation is likely not governed largely by collective cell properties but rather by individual cell properties. This conclusion agrees with the fact that tumor cells often downregulate cell–cell adhesion molecules such as E-cadherin [55] and thus reduce mechanical coupling with each other.

## 5. Conclusions

The current study confirmed the pro-proliferative effect of MACC1 [5,39,43,56,57] and found that differences in migration between high- and low-MACC1-expressing cells diminished in the absence of proliferation. Thus, we conclude that proliferation is the main driver of differential migration in high- and low-MACC1-expressing cells here. Consequently, it could be argued that targeting MACC1-induced proliferation may affect migration as well.

## Figures and Tables

**Figure 1 cancers-14-02857-f001:**
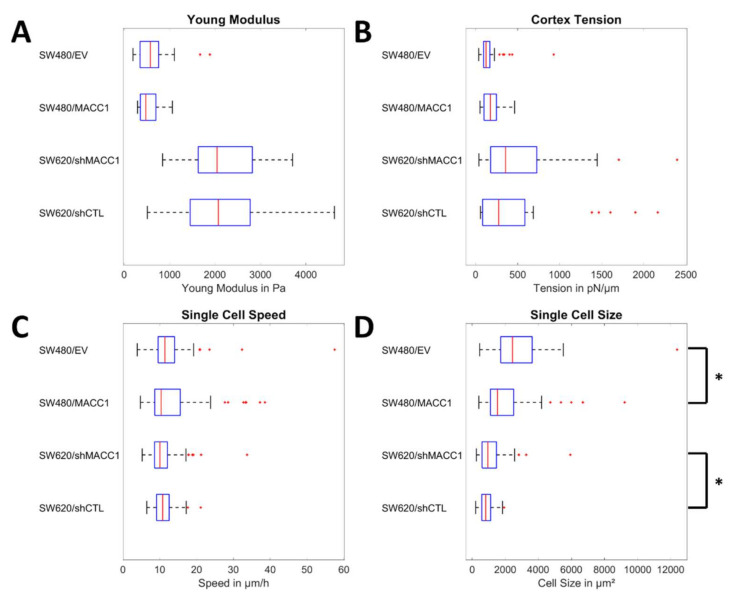
Single cell properties of high- and low-MACC1-expressing colon carcinoma cells. (**A**,**B**) depict the results of the biomechanical measurements for the Young´s modulus and the cortex tension. (**C**,**D**) show the results of live cell imaging of single cells for the mean speed and the contact area with the substrate. Sample sizes: (**A**) n_SW480/EV_ = 35; n_SW480/MACC1_ = 33; n_SW620/shMACC1_ = 40; n_SW620/shCTL_ = 40. (**B**) n_SW480/EV_ = 33; n_SW480/MACC1_ = 31; n_SW620/shMACC1_ = 25; n_SW620/shCTL_ = 26. (**C**,**D**) n_SW480/EV_ = 66; n_SW480/MACC1_ = 98; n_SW620/shMACC1_ = 102; n_SW620/shCTL_ = 111. Asterisk depicts statistically significant results with *p* < 0.05. Box plots show the median (red line), 25 and 75 percentile (box), non-outlier range (whiskers), and outliers (red dots).

**Figure 2 cancers-14-02857-f002:**
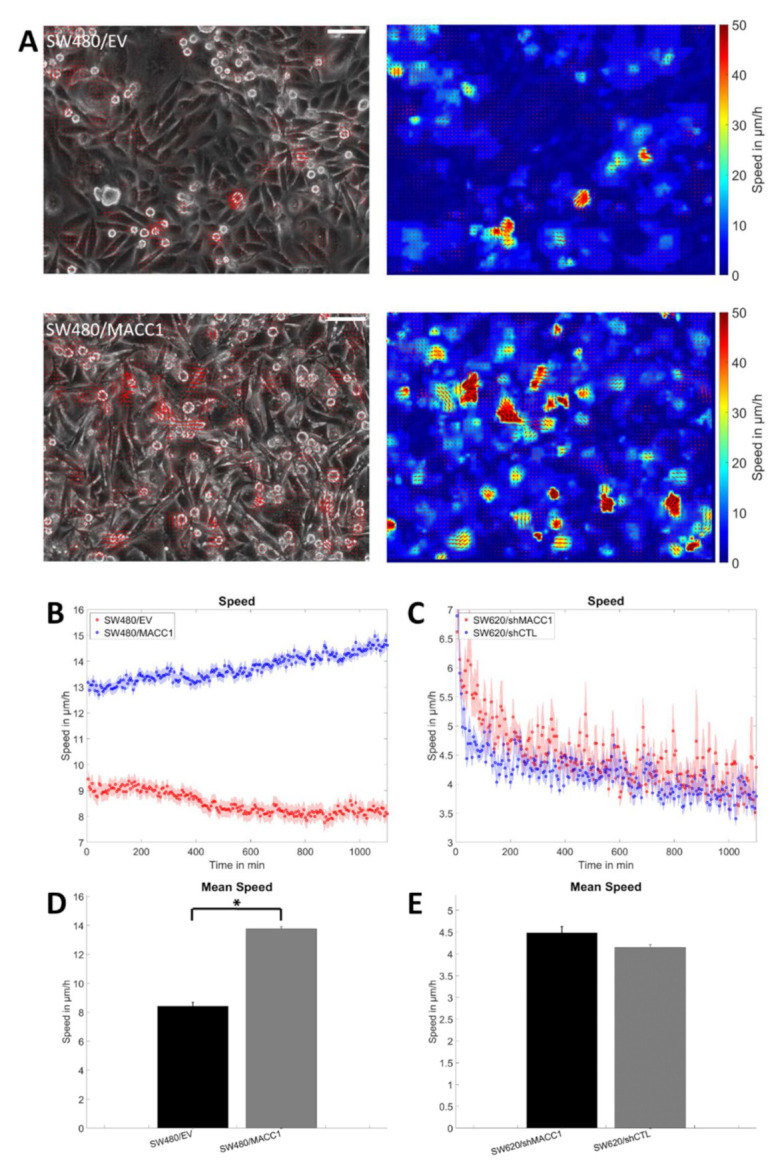
Collective migration of SW480 and SW620 cells. (**A**) The left column shows a typical phase contrast image of SW480, overlaid with the vectors of the velocity fields, and the right column shows the magnitude of the velocity and its direction. The scale bar depicts 50 µm. (**B**,**C**) Graph of the mean speeds of SW480 and SW620 cells, respectively. (**D**,**E**) Mean of the movement speeds of SW480 and SW620 cells from data in (**B**,**C**). Sample sizes: n_SW480/EV_ = 75; n_SW480/MACC1_ = 75; n_SW620/shMACC1_ = 32; n_SW620/shCTL_ = 32. Asterisk depicts statistically significant results with *p* < 0.05. Error bars and shaded areas depict the standard error of the mean.

**Figure 3 cancers-14-02857-f003:**
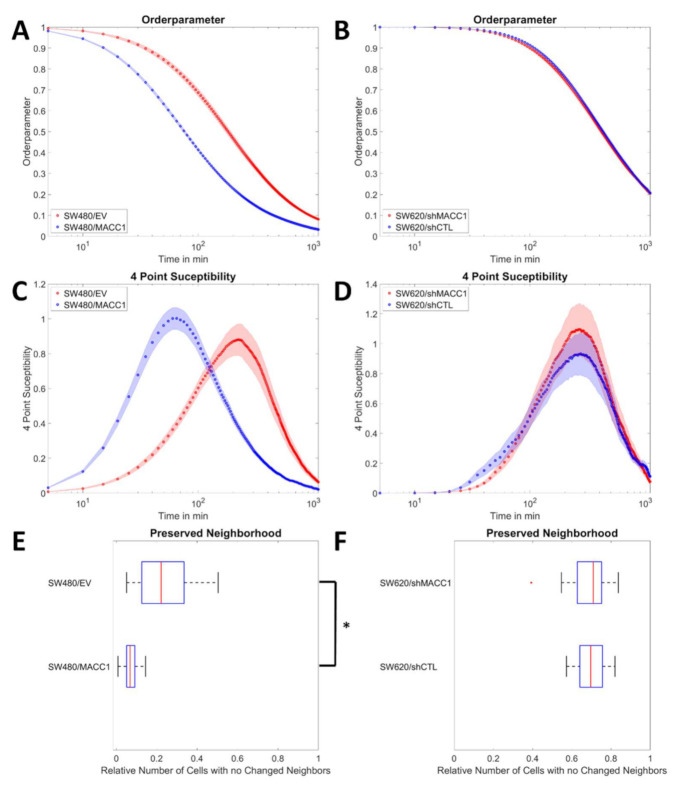
Collective migration properties of SW420 and SW620 cells. (**A**,**B**) Order parameter Q for both cell lines for the 20 h time window. (**C**,**D**) Four-point susceptibility as obtained from the velocity fields over the whole measurement time. The peak positions of the 4-point susceptibility represent the average life time of collectively moving packs of cells. (**E**,**F**) Analysis of the layer reorganization in terms of cells that did not make new neighbors during live-cell imaging. Sample sizes: n_SW480/EV_ = 75; n_SW480/MACC1_ = 75; n_SW620/shMACC1_ = 32; n_SW620/shCTL_ = 32. Shaded areas depict the standard error of the mean. Box plots show the median (red line), 25 and 75 percentile (box), non-outlier range (whiskers), and outliers (red dots). Asterisk depicts statistically significant results with *p* < 0.05.

**Figure 4 cancers-14-02857-f004:**
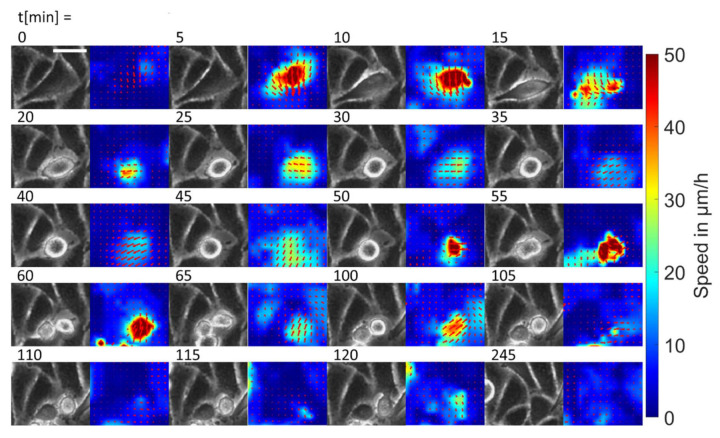
Effects of cell division on local cellular velocities. This image collection depicts a single cell division, together with the associated local speeds as a function of time. Please denote the high speeds during mitotic rounding and subsequent high speeds during the expansion of the two daughter cells. The scale bar depicts 50 µm.

**Figure 5 cancers-14-02857-f005:**
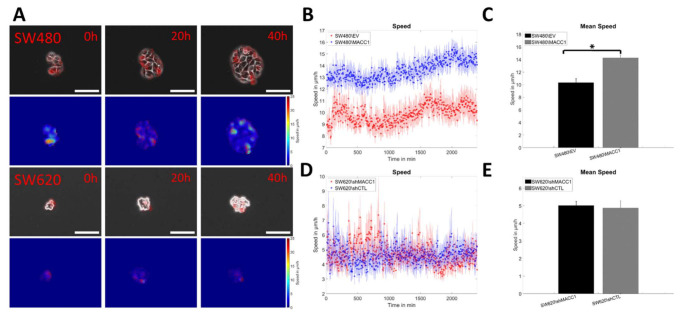
Expansion of small cell clusters. (**A**) Illustration of cluster expansion for SW480 and SW620 cells during the measurement time, together with the associated velocity maps. The scale bar depicts 50 µm. (**B**,**D**) Plots of the cellular speed inside the clusters as a function of time for both cell types. (**C**,**E**) Temporal averages of the cell speed inside the clusters for SW480 and SW620 cells. Sample sizes: n_SW480/EV_ = 14; n_SW480/MACC1_ = 14; n_SW620/shMACC1_ = 18; n_SW620/shCTL_ = 17. Asterisk depicts statistically significant results with *p* < 0.05. Error bars and shaded areas depict the standard error of the mean.

**Figure 6 cancers-14-02857-f006:**
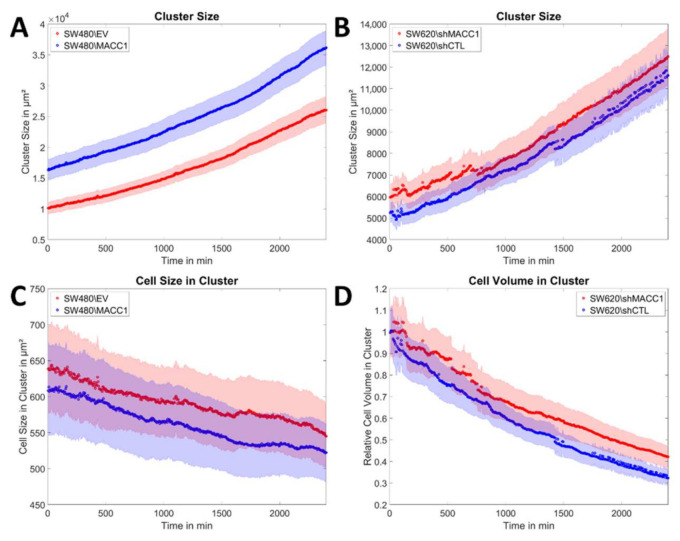
Evolution of cluster and cell size as a function of time. (**A**,**B**) Average size of cell clusters of SW480 and SW620 cells as a function of time. (**C**) Cluster size of SW480 cells normalized to the cell number, corresponding to the average cell size. (**D**) Cluster size of SW620 cells normalized to the average cell volume at time point 0. Sample sizes: n_SW480/EV_ = 14; n_SW480/MACC1_ = 14; n_SW620/shMACC1_ = 18; n_SW620/shCTL_ = 17. Shaded areas depict the standard error of the mean.

**Figure 7 cancers-14-02857-f007:**
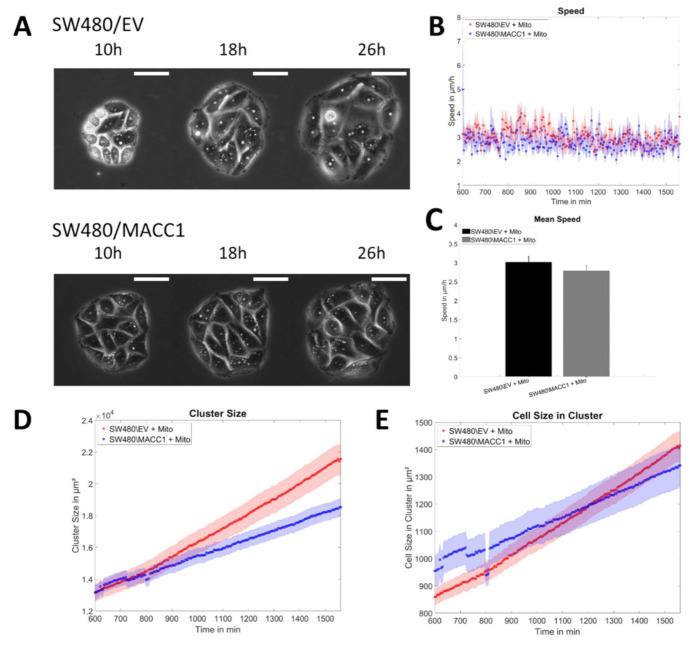
Expansion of small cell clusters without proliferation. (**A**) Illustration of cluster expansion for SW480 cells treated with mitomycin (Mito) during the measurement time, without proliferation. The scale bar depicts 50 µm. (**B**,**C**) Speed of cells in clusters of SW480 cells as a function of time or averaged over time. (**D**,**E**) Total cluster size of SW480 cells over time and normalized to the cell number. Sample sizes: n_SW480/EV+Mito_ = 18; n_SW480/MACC1+Mito_ = 15. Error bars and shaded areas depict the standard error of the mean.

## Data Availability

All data is contained within the manuscript. Source codes are available from the authors on request.

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
