# Peer review of "MACC1-Induced Collective Migration Is Promoted by Proliferation Rather Than Single Cell Biomechanics"

_cancers, 2022, doi:10.3390/cancers14122857_

Round 1

Reviewer 1 Report

In the manuscript entitled “MACC1 induced collective migration is promoted by proliferation rather than single cell biomechanics”, Hohmann et al. analyze the role of MACC1 in regulating biomechanical and migratory properties of colorectal cancer (CRC) cells. Despite the fact that the manuscript presents a good experimental design, the manuscript needs to be revised and integrated with additional data in order to improve and give more validity to the manuscript.

  • For all the experiments, authors used two established cell lines, expressing low (SW480) and high (SW620) MACC1 levels. To improve the robustness of their findings, authors should perform experiments with other CRC cells expressing low and high MACC1 endogenous levels.
  • Moreover, to increase the translational significance of the data, authors should validate data with a direct/indirect inhibitor of MACC1.
  • Authors should include the scale bar in all the images.
  • Authors should carefully checked the number of figures during the manuscript. In the 3.3 result section, authors described data reported in Figure 5, but in the figure legend this figure is indicated as Figure 1. The same discrepancy was repeated for figures 6 and 7.

Author Response

In the manuscript entitled “MACC1 induced collective migration is promoted by proliferation rather than single cell biomechanics”, Hohmann et al. analyze the role of MACC1 in regulating biomechanical and migratory properties of colorectal cancer (CRC) cells. Despite the fact that the manuscript presents a good experimental design, the manuscript needs to be revised and integrated with additional data in order to improve and give more validity to the manuscript.

1) For all the experiments, authors used two established cell lines, expressing low (SW480) and high (SW620) MACC1 levels. To improve the robustness of their findings, authors should perform experiments with other CRC cells expressing low and high MACC1 endogenous levels.

2) Moreover, to increase the translational significance of the data, authors should validate data with a direct/indirect inhibitor of MACC1.

We are thankful for these suggestions to improve the manuscript and further broaden the outcome of the work. To Points 1) and 2): We would like to point out that for all experiments 2 different cell lines, namely SW620 and SW480, were used, not only in control conditions, but also stably transfected to either express high amounts of MACC1 (SW480/MACC1) or low amounts of MACC1 (SW620/shMACC1). Consequently, we considered the more specific regulation of MACC1 more suitable than pharmacological intervention, especially if the underlying signaling cascade and off-targets of inhibitors are not well known. 

Furthermore, additional experiments could not be conducted in the given period of 10 days for this “major revision”. For illustration, cells need to be thawed, cultured and validated regarding MACC1 expression, taking at least 2 weeks alone. Before starting experiments additional validation experiments for cell density, cellular dynamics, necessary mitomycin concentration, etc. need to be performed and analyzed, yielding image numbers in the order of 10,000 images, necessitating at least 2-3 weeks for performing and analyzing images. Afterwards, experiments need to be done, including single cell migration experiments (1d measurement time per repetition), collective migration experiments (1d measurement time per repetition) and cluster expansion experiments (2d measurement time per repetition). If all experiments are performed, they will yield, depending on the necessary settings to capture cellular dynamics, at least 50,000 images that need to be analyzed. Overall, it adds up to several month of additional work. Consequently, we were unfortunately not able to perform additional experiments during the given short time period of this revision.

3) Authors should include the scale bar in all the images.

We thank the reviewer for this valuable hint. Scale bars were added to all phase contrast images in figure 2, 5, 7 and S4.

4) Authors should carefully checked the number of figures during the manuscript. In the 3.3 result section, authors described data reported in Figure 5, but in the figure legend this figure is indicated as Figure 1. The same discrepancy was repeated for figures 6 and 7.

We thank the reviewer for pointing out this issue. The respective figures have been labeled correctly. We apologize for any confusion.

Reviewer 2 Report

Methods used to study cell motility and colony expansion  are interesting.

Add more recent references.

1) The research aims to clarify whether biomechanics and single cell proliferation interfere with MACC1-dependent migration.  

2) I consider the topic original since this paper summarizes important observations about the role of MACC1 dysregulation in cancer, that is still poorly understood.  

3) Compared with other publications, authors come to the conclusion for the first time that MACC1 overexpression affects only the collective migration of cells linked to proliferation, but not the biomechanics of single cells.  

4) Description of procedures needs minor clarification.  

5) Conclusions are clearly supported by data and are linked to goals.  

6) References are appropriate to article type. Authors are recommended to add recent articles.  

7) Data in tables and figures support the text. 

Author Response

Methods used to study cell motility and colony expansion  are interesting.

Add more recent references.

More recent references were added.

1) The research aims to clarify whether biomechanics and single cell proliferation interfere with MACC1-dependent migration.  

2) I consider the topic original since this paper summarizes important observations about the role of MACC1 dysregulation in cancer, that is still poorly understood.  

3) Compared with other publications, authors come to the conclusion for the first time that MACC1 overexpression affects only the collective migration of cells linked to proliferation, but not the biomechanics of single cells.  

We thank the reviewer for the encouraging feedback.

4) Description of procedures needs minor clarification.  

We thank the reviewer for this suggestion. Further explanations were added to the description of the atomic force microscopy measurements.

5) Conclusions are clearly supported by data and are linked to goals.  

We thank the reviewer for this evaluation.

6) References are appropriate to article type. Authors are recommended to add recent articles.  

We thank the reviewer for the advice. Older citations have been replaced by new ones.

7) Data in tables and figures support the text. 

We thank the reviewer for this statement.

Reviewer 3 Report

Using CRC cell lines, the authors investigate the properties of cells with high MACC1 expression to determine the influence of MACC1 on cell motility and intracellular dynamics.

Overall the findings support the conclusion however there are some issues that I believe must be addressed.

Additionally, the English grammar is often incorrect or clumsy. I suggest a further edit by a native English speaker.

See suggests below.

In the Material and Methods section, the authors apparently “cut and pasted” the method for atomic force microscopy because they describe the experiments done in ref 13 and not the experiments done in this paper. (Ln 90).

Ln 106 – please describe briefly how the experiment was done.

Ln 181 – the authors describe the cell lines with high, low and endogenous expression of MACC1 and refer us to a previous publication. I think that we should be able to assess expression of MACC1 now in these cell lines. I suggest a western blot that can be in the supplemental figures.

Figure 1 – It would be helpful to add the n-value (how many single cells were measured for each condition either in the figure or in the figure legend. (for all the panels). Additionally, in the text describing this figure – ln 185, ln 188, ln 195, the authors state that the findings are significant. Please provide a p-value for this is the text so we can assess significance specifically for each of these measure (and not generally as stated in the legend that * is p<0.05)

The n-value and p-value should be stated throughout the paper (n-value in the figure legends and p-value whenever “significance” is claimed in the text.

Figure legends 5, 6, 7 are mislabeled.

In Figure 5, the pictures of the cells in A are informative and give the reader a feeling for the description provided by the authors of the cell behavior. Thus, it would be even more informative if this panel were larger and could even include another example of each cell type – while panel C and E could be smaller without losing any information.

Author Response

Using CRC cell lines, the authors investigate the properties of cells with high MACC1 expression to determine the influence of MACC1 on cell motility and intracellular dynamics.

Overall the findings support the conclusion however there are some issues that I believe must be addressed.

Additionally, the English grammar is often incorrect or clumsy. I suggest a further edit by a native English speaker.

We re-evaluated the manuscript to improve readability and correct errors.

See suggests below.

1) In the Material and Methods section, the authors apparently “cut and pasted” the method for atomic force microscopy because they describe the experiments done in ref 13 and not the experiments done in this paper. (Ln 90).

We thank the reviewer for pointing out this issue. We rephrased the respective part and explained the Young’s modulus measurements in more detail. Yet, we have to denote, that a part of the measurements presented here were performed very similar as in the cited study, performed previously in our lab (calculation of Young’s modulus).

We rephrased the referred part, also adding details about the disentanglement of both quantities measured with the AFM, as asked in point 2. Underlined parts were added or modified. It now reads as follows:

Mechanical properties of single cells were assessed in the form of the Young’s modulus and cortex tension; both measured using an atomic force microscope (AFM; Bruker, Billerica, MA, USA, Bioscope Catalyst). The measurement procedure for obtaining the Young’s modulus is described elsewhere 1,2. Briefly, cells were seeded on a petri dish and measured 15 min after seeding, to avoid slippage of individual cells. Measurements were conducted using a tip-less cantilever (Arrow-TL2, Nanoworld, Neuchatel, Switzerland) applying a force of 1 nN that leads to deformations of 1-2 µm. The Young’s modulus was calculated using the Hertz model (please see attached file):

Here, F denotes the applied force, E the Young’s modulus, R the cells radius, n the Poisson ratio (set to 0.5) and d0 the central indentation.

From the same measurement curves, the actin cortex tension was extracted. For this purpose, a model introduced by Cartagena-Rivera et al 3,4 was used (please see attached file):

Here T is the cortex tension, k the elastic constant of the cantilever, Z the piezo extension and d the deflection of the cantilever. For calculation of the surface tension the first 200 nm of the force-distance curve after contact between cantilever and cell were fitted. Please denote, that the difference in indentations (factor 5-10) allows a disentanglement between cortical Young’s modulus and cortical tension. When fitting force distance curves for extraction of either the Young’s modulus or the cortex tension, values were discarded if R²<0.8.

2) Ln 106 – please describe briefly how the experiment was done.

We would kindly refer to the answer of point 1 as well. The data for the cortex tension was extracted from the same measurement curves as the values for the Young’s modulus. For calculating the cortex tensions only the first 200nm of the force indentation curve were used, after the cantilever made contact with the cell. The Young´s modulus was obtained by indenting a monolayer with 1nN leading to indentations of about 1000-2000nm. Thus, indentation were at least lower by a factor 5 for cortex tension extractions, allowing the disentanglement of cortex tension and Young’s modulus.

Furthermore, for the measurements cells that just started to adhere were used. Those cells do not contain an intact cytoskeleton, including most actin structures, except for the actin cortex, additionally facilitating the disentanglement of cortex tension and elasticity 5–7.

3) Ln 181 – the authors describe the cell lines with high, low and endogenous expression of MACC1 and refer us to a previous publication. I think that we should be able to assess expression of MACC1 now in these cell lines. I suggest a western blot that can be in the supplemental figures.

We thank the reviewer for the advice. mRNA expression data and western blot have been added.

4) Figure 1 – It would be helpful to add the n-value (how many single cells were measured for each condition either in the figure or in the figure legend. (for all the panels). Additionally, in the text describing this figure – ln 185, ln 188, ln 195, the authors state that the findings are significant. Please provide a p-value for this is the text so we can assess significance specifically for each of these measure (and not generally as stated in the legend that * is p<0.05)

The n-value and p-value should be stated throughout the paper (n-value in the figure legends and p-value whenever “significance” is claimed in the text.

We thank the reviewer for this suggestion. All sample sizes were summarized in supplemental table 1. Furthermore, sample sizes were added into the figure legends. p-Values were included to the main text.

5) Figure legends 5, 6, 7 are mislabeled.

We thank the reviewer for pointing out this issue. The respective figures have been labeled correctly.

6) In Figure 5, the pictures of the cells in A are informative and give the reader a feeling for the description provided by the authors of the cell behavior. Thus, it would be even more informative if this panel were larger and could even include another example of each cell type – while panel C and E could be smaller without losing any information

We thank the reviewer for the advice. Videos of cluster expansion, including the detected edges for SW480/EV, SW480/MACC1, SW620/shCTL and SW620/shMACC1 were added to the supplement, because we considered these highly instructive for understanding the underlying process.

Bibliography:

  1. Hohmann, T., Grabiec, U., Ghadban, C., Feese, K. & Dehghani, F. The Influence of Biomechanical Properties and Cannabinoids on Tumor Invasion. Cell Adh. Migr. 11, 54–67 (2017).
  2. Hohmann, T. & Dehghani, F. Measuring Mechanical and Adhesive Properties of Single Cells Using an Atomic Force Microscope. in Metastasis: Methods and Protocols (ed. Stein, U. S.) 81–92 (Springer US, 2021). doi:10.1007/978-1-0716-1350-4_6.
  3. Cartagena-Rivera, A. X., Logue, J. S., Waterman, C. M. & Chadwick, R. S. Actomyosin Cortical Mechanical Properties in Nonadherent Cells Determined by Atomic Force Microscopy. Biophys. J. 110, 2528–2539 (2016).
  4. Logue, J. S. et al. Erk regulation of actin capping and bundling by Eps8 promotes cortex tension and leader bleb- based migration. Elife (2015).
  5. Prahlad, V., Yoon, M., Moir, R. D., Vale, R. D. & Goldman, R. D. Rapid movements of vimentin on microtubule tracks: Kinesin-dependent assembly of intermediate filament networks. J. Cell Biol. 143, 159–170 (1998).
  6. Guck, J. et al. Optical deformability as an inherent cell marker for testing malignant transformation and metastatic competence. Biophys. J. 88, 3689–3698 (2005).
  7. Bereiter-Hahn, J., Lück, M., Miebach, T., Stelzer, H. K. & Vöth, M. Spreading of trypsinized cells: cytoskeletal dynamics and energy requirements. J. Cell Sci. 96 ( Pt 1), 171–88 (1990).

Round 2

Reviewer 1 Report

The authors have satisfactorily addressed most of my concerns. Additional details have improved the clarity of the manuscript.